# Getting Bored of Cyberwar: Exploring the Role of Low-level Cybercrime Actors in the Russia-Ukraine Conflict

## ABSTRACT

There has been substantial commentary on the role of cyberattacks carried by low-level cybercrime actors in the Russia-Ukraine conflict. We analyse 358k web defacement attacks, 1.7M reflected DDoS attacks, 1 764 Hack Forums posts mentioning the two countries, and 441 announcements (with 58k replies) of a volunteer hacking group for two months before and four months after the invasion. We find the conflict briefly but notably caught the attention of low-level cybercrime actors, with significant increases in online discussion and both types of attack targeting Russia and Ukraine. However, there was little evidence of high-profile actions; the role of these players in the ongoing hybrid warfare is minor, and they should be separated from persistent and motivated 'hacktivists' in state-sponsored operations. Their involvement in the conflict appears to have been short-lived and fleeting, with a clear loss of interest in discussing the situation and carrying out both defacement and DDoS attacks against either Russia or Ukraine after a few weeks.

## CCS CONCEPTS

• **Social and professional topics** → **Computer crime**; • **Applied computing** → **Cyberwarfare**; • **Security and privacy** → Social aspects of security and privacy; • **Mathematics of computing** → Time series analysis; • **Networks** → Denial-of-service attacks.

## KEYWORDS

DDoS attacks; web defacements; Russia-Ukraine conflict; low-level cybercrime; cyberwar; volunteer hacktivists; IT Army of Ukraine.

## 1 INTRODUCTION

Researchers, politicians, and journalists have long been fascinated by 'cyberwar' – the spectre of armed conflict between nations spilling over into attacks conducted over the Internet [59]. 'Colder' forms of inter-state conflict are characterised by espionage and intelligence gathering, which may facilitate the degradation of online systems once hostilities commence [16]. Alongside this there has been a thirty-year history of speculation around how the tools and techniques of the cybercrime underground – Distributed Denial of Service (DDoS) attacks, disruption and compromise of services, web defacements, and similar techniques – might allow civilians to play a role in a 'hot' war between developed nations [5]. Much of this speculation, drawing from criminological models of low-level cybercrime groups and on links between this underground and well-organised 'hacktivist' movements, has argued these groups would play a crucial role, making the future of war hybrid, chaotic, and unpredictable [74]. The 2022 invasion of Ukraine provides an opportunity to assess what has happened in practice.

Russia and Ukraine have a long history of electronic information warfare [31] and are among the most active cybercrime hubs [40]. When Russia invaded Ukraine on 24 February 2022, war-related attacks hitting the two countries were regularly reported [12]. A popular narrative is that the engagement of low-level cybercrime actors and volunteers could be a game changer and could undermine Russia's war [56]. Some commentators predicted it will be the first full-scale cyberwar [13], its effects will last for decades [61], and youngsters would be drawn into a 'cyberwar' by joining IT Army of Ukraine – a group backed by the Ukrainian state to co-ordinate volunteers and civilians to help disrupt Russian assets [23, 63]. Some have suggested a real cyber war, predicting hacktivist attacks on Russia would escalate further throughout 2022 [2]. These narratives regularly appear in the press and play a role in shaping domestic policy responses to cybercrime. Although less likely to grab headlines, a contrary narrative around 'overhyped cyberwar' suggests cyber operations in the conflict have been too slow [42], surprisingly insignificant [34], while the unprecedented level of cyberattacks and Russia's vaunted cyber capabilities are questionable [18, 19, 77]. GCHQ commented the cyber conflict had not yet materialised [17] and pointed to the resilience of Ukraine's defences [69].

Government-backed cyber operations [21, 43] and destructive crimes have continued [78]. Yet, data about nation-state attacks is hard for academics to access, and those behind significant real-world attacks tend to take steps to avoid scrutiny. We are particularly interested in non-governmental activity contributed by many low-level but high-volume actors, focusing on the hypothetical 'volunteer army', where participants are mostly unskilled and their activity highly relies on off-the-shelf tools. We explore their role in the 'cyberwar' between Russia and Ukraine, in which both sides have substantial IT infrastructure, a thriving digital underground crime ecosystem, and significant access to offensive capacities.

We longitudinally and statistically measure activities linked with low-level cybercrime actors, including web defacements (§4) and DDoS attacks (§5). The findings are incorporated with analyses of discussions of the general hacking community, and a pro-Ukraine volunteer group (§6). The role of these low-level actors in the conflict is discussed in §7. Our study was approved by our institutional Research Ethics Board (Appendix §A). All data and scripts are available for academic researchers on request (Appendix §B).

## 2 BACKGROUND AND RELATED WORK

Information warfare has long been a routine part of 'hybrid' modern conflicts, especially around controlling communications [25, 38]. The enemy's capability to spread news and propaganda can be impacted by targeting crucial sites, public services, broadcast and telecom infrastructure. Censorship is often used during wartime [57]; governments block access to global services, especially social networks and media platforms to suppress unwanted narratives. Russia blocked news and anti-war domains when the conflict started [58, 65], and lost access to foreign service providers [32] and websites [58]. Ukrainian users experienced degraded network performance [30], while pro-Ukrainian supporters have tried unconventional channels such as online reviews to bypass censorship [45].

Table 1: The complete collection of five most popular defacement archives for 6 months from 1 January 2022 to 30 June 2022.

| | Zone-H zone-h.org | OwnzYou ownzyou.com | Zone-Xsec zone-xsec.com | Haxor-ID hax.or.id | Defacer-Pro defacer.pro | Total 5 archives |
|---|---|---|---|---|---|---|
| Archive URL | | | | | | |
| Manual staff verification | ✓ | · | · | · | · | ✓ |
| Automatic validity sanitisation | · | · | ✓ | ✓ | ✓ | ✓ |
| Team information | · | · | ✓ | ✓ | ✓ | ✓ |
| Country of targeted victims | ✓ | ✓ | ✓ | ✓ | ✓ | ✓ |
| Originating country of defacers | · | · | · | ✓ | · | ✓ |
| Reason and motivation | · | · | · | ✓ | · | ✓ |
| Type of vulnerability | · | · | · | ✓ | · | ✓ |
| Snapshots of defaced websites | ✓ | ✓ | ✓ | ✓ | ✓ | ✓ |
| Defacements (raw) | 164 312 | 76 608 | 53 852 | 34 482 | 28 594 | 357 848 |
| Defacements [†] | 164 312 | 67 510 | 53 814 | 34 465 | 27 662 | 317 049 |
| Valid defacements [†] | 143 485 (87.32%) | 47 657 (70.59%) | 53 705 (99.80%) | 34 439 (99.92%) | 26 379 (95.36%) | 274 963 (86.73%) |
| Invalid defacement [†] | 20 827 (12.68%) | 19 853 (29.41%) | 109 (0.20%) | 26 (0.08%) | 1 283 (4.64%) | 42 086 (13.27%) |
| Defacers (raw) | 2 173 | 1 214 | 561 | 484 | 540 | 4 347 |
| Defacers [†] | 1 790 | 689 | 560 | 482 | 526 | 3 454 |
| Defacers with valid reports [†] | 1 655 (82.01%) | 553 (54.00%) | 541 (99.82%) | 443 (99.55%) | 486 (97.79%) | 2 781 (77.44%) |
| Defacers with invalid reports [†] | 843 (41.77%) | 722 (70.51%) | 24 (4.43%) | 15 (3.37%) | 147 (29.58%) | 1 656 (46.12%) |

✓ fully available; ✓ partly available; · not available; [†] duplicated defacements and defacer handles within and across different archives were unified.

Some links between kinetic warfare and 'nationalistic' cyberattacks has been reported. Ukrainian firms were hit by data wipers such as CaddyWiper and NotPetya [1, 49], DDoS attacks [8, 64] and phishing campaigns [36]; Ukraine supporters have used spam senders to distribute propaganda in Russia [71] and have stolen cryptocurrency from Russian wallets [73]. Ukrainian universities were hacked [80], the Ukrainian electricity grid was hit by Industroyer2 [24], and the Ukrainian satellite Internet was downed by Russia [53]. Attackers identifying themselves under the banner of the Anonymous movement declared a 'cyberwar' on Russia [44] with attacks against Russian Ministry of Defence databases [37] and state TV channels [48]. Russia intermittently received attacks instigated by volunteer hacktivists of the IT Army of Ukraine [10, 56].

While the security industry has reported some insights [21, 43, 46, 47], empirical quantitative academic work analysing the link between armed conflicts and cybercrime has been limited. A notable report is by a Czech university's incident response team, showing negligible impact on their network after hundreds of users launched DDoS attacks against Russia for a week after the invasion [28].

One type of attack linked with the low-level cybercrime actors is web defacement [60], which accounted for around 20% of online attacks in 2014 [50] and is often organised into discrete campaigns [41]. Proactive defacements can evade URL safety-checking tools [81]. Attackers (or defacers) gain unauthorised access using off-the-shelf tools and simple exploits, then alter sites' appearance to demonstrate success [41]. Defacers have heterogeneous developmental trajectories [72]; they are often organised in groups [54] and have been using online archives [35] as a 'hall of fame' to show off their achievements to gain reputation. Defacements are mostly hobbies or pranks with greetings to peers [79], but some advertise tools or hacking services to make money, or express other motives such as a wish for community recognition, patriotic, religious and political views [6, 60]. Defacement may cause economic harm [4, 15] and has occasionally been used as a proxy for terrorist and other serious activities [26]. Another simple type of large-scale attack associated with low-level actors is amplified DDoS. DDoS-as-a-service sites abound [33], and off-the-shelf DDoS tools are widely available; they were tailored and provided to pro-Ukrainian volunteers early in the conflict to attack Russian infrastructure.

Unlike state-sponsored activities, defacement and DDoS attacks can be systematically collected and measured with reasonable completeness. Defacements are available on online archives [35], while DDoS attacks can be collected through honeypots [70]. Launching these attacks with ready-made tools is straightforward for those without much technical expertise. They can be executed quickly, at scale, and have instant, noticeable effects such as altering targets' appearance, making them inaccessible, or taunting opponents with compromised sites. During wartime, the need to rapidly disseminate political messages and propaganda makes them attractive.

## 3 METHODS AND DATASETS

We use several quantitative datasets collected regularly and separately, spaning 1 January to 30 June 2022; timestamps are normalised to UTC. To determine if the conflict has impacts resulting in different means (or mean ranks) of daily cyberattacks and hacking discussions, we separate the period into three eras; $E_1$: before the invasion, from 1 January to 24 February 2022; $E_2$: around one month immediately after the invasion, from 24 February to 31 March 2022; and $E_3$: from 1 April to 30 June 2022. We then apply unpaired statistical tests, using One-way ANOVA or Kruskal-Wallis depending on the data distribution; the null hypothesis $H_0$ is there is no significant difference between three eras. We use post-hoc tests Tukey-Kramer for ANOVA or Dunn's for Kruskal-Wallis to identify pairs causing the changes if any. The effect size is measured by $\eta^2$, ranging [0, 1]; $0 \leq \eta^2 < 0.01$: almost no effect; $0.01 \leq \eta^2 < 0.06$: small effect; $0.06 \leq \eta^2 < 0.14$: medium effect; $0.14 \leq \eta^2 \leq 1$: large effect [62].

**Web Defacement Attacks.** We fully scrape the most trusted active defacement archives during the period, see Table 1. The largest and

most popular archive is Zone-H (since March 2002), others similar ones include OwnzYou (since January 2021), Zone-Xsec (since May 2020), Haxor-ID (since November 2019), and Defacer-Pro (since June 2021). Smaller archives were historically active [41], but either vanished (Hack Mirror and Mirror Zone) or have hosted different content (Hack-CN and MyDeface). While not all compromised sites get reported, measuring trends from the most reputed archives is likely informative. The country of defaced sites is identified based on ccTLD, IP geolocation, and geolocation of the AS hosting the sites, excluding CDNs (Appendix §C). The defacement submission process is detailed in Appendix §D. We ensure data completeness and bypass challenges e.g., Captcha and IP blocking (Appendix §E).

Further steps are performed to enhance data reliability. First, many on-hold submissions are valid but were never verified; we perform a semi-automatic validation using the messages left on defaced pages (Appendix §F). Second, submissions may be reported to multiple archives to broaden their visibility. We de-duplicate across and within archives by hashing their content (see Appendix §G). Third, as 'notifier' can be arbitrary, typos can give a single attacker multiple identities; we correct typos across all archives using handles' similarity and messages left on defaced pages (see Appendix §G). In total, 137 339 reports were verified by the archives, 97 652 were automatically validated by us and a further 39 972 were validated semi-automatically. 40 799 (11.00%) duplicate reports are merged across all archives. Of the remaining 317 049 reports, we analyse the 274 963 validated submissions (86.73%, around 1 500 per day). Of these, 4 347 defacer handles are also unified to 3 454.

**UDP Amplification DDoS Attacks.** We use 1.7M DDoS attack records gathered by a honeypot network emulating protocols vulnerable to reflected UDP attacks [70]. A flow of packets is considered to be an attack if any sensor observes at least 5 packets for the same victim IP or IP prefix, and the attack is deemed to last from the first packet until the last packet preceding a 15 minute period without further packets. In 2022, the median number of honeypots contributing data was 50, 95% CI [34, 51]; the median number of observed attacks per week was 35 000, 95% CI [11 900, 271 000] and on IP prefixes of 438, 95% CI [0, 3 480]; the median attack duration was 1.53 minutes, while the maximum was 11 300 minutes. The country of victims is identified based on IP geolocation and geolocation of the AS hosting that IP, excluding CDNs (see Appendix §C).

**Underground Forum Discussions.** Online forums are structured around subforums containing threads with multiple posts. To assess the changes of discussion within the hacking community, we use a snapshot of the most popular hacking forum, Hack Forums from the CrimeBB dataset [52]. The forum is a place for users to learn about attacks and trade in cybercrime tools and services. Many are low-level actors, however some have been prosecuted for cybercrime-related activities [51]. We extract all 123 threads within the 6-month period consisting of at least one post with the keywords 'Russia' and/or 'Ukraine' (case-insensitive): 115 related to Russia, 108 related to Ukraine, in which 100 related to both. We then use all 1 279 posts from 84 highly relevant threads – those with titles directly having the keywords. For the rest 39 less-relevant threads, we count 485 posts directly consisting of the keywords. In total, 1 764 relevant posts made by 372 users are analysed.

**Volunteer Hacking Discussions.** Two days after the invasion, the Ukrainian government called on pro-Ukraine 'hacktivists' to join the IT Army of Ukraine, which was stood up in an ad-hoc manner [63] to support the war effort [23, 67]. The most tangible outcome is a public Telegram channel mainly used to recruit and encourage volunteers to spread news, propaganda, and co-ordinate disruptive efforts against Russia. We confirmed with a Ukrainian government source that it is the official channel used for communication amongst Ukrainian civilians, with messages being forwarded to other unofficial satellite groups with far fewer subscribers.

The group, attracting more than 200k subscribers, promotes lists of Russian targets (both in Ukrainian and English) most mornings with URLs and IP addresses posted on their Telegram channels. They encourage using various attack vectors to disrupt communication and financial systems by hitting banks, businesses, government, and logistics [10]. They provide guides and tools for launching attacks e.g., tools for quickly fetching daily targets and granting access to individuals' cloud resources for later coordinated attacks. This 'cyber army' claims ordinary Russians have seen impacts when they hit banks, exchanges [7], and cinemas [55].

We believe the involved 'volunteer hacktivists' are mostly low-level actors, as much of their activity depends on tools provided by the group. We collect 441 announcements with 57 757 replies and 900k emoji reactions posted in the channel from its inception until 30 June 2022 using Telethon, which interacts with official Telegram APIs to fully capture messages and metadata. We then used regular expressions to extract promoted IP addresses and domains; subdomains such as www.xyz.ru and smtp.xyz.ru are combined. Besides Russian and Belarusian domains (.ru, .su, .by), top level domains (e.g., .tv, .com) are also targeted. URL shorteners (e.g., goo.gl) and online services (e.g., youtube.com) are excluded, resulting in 3 845 targets: 2 291 IP addresses (59.58%) and 1 554 domains (40.42%).

## 4 THE EVIDENCE FROM WEB DEFACEMENTS

We measure the dynamic of defacements, both on the global and Russia-Ukraine scales. Figure 1 shows the number of defacements per day as the conflict progressed. Figure 2 breaks down changes by hour for the most active four-week period from 17 February 2022. **The Russia-Ukraine Scale.** The number of defacements targeting Russia immediately peaked on the invasion day at 209 (14.48% of all defacements on that day, while it was 0.60% the day before). The first big wave was at around 10AM (7 hours after the invasion) with 178 attacks caused by a single defacer, followed by smaller waves on the same day. Two follow-up waves occurred at 1PM on 25 February and 9AM on 26 February with 43 and 109 attacks, respectively. The number of defacers targeting Russia peaked 2 days later: while only 11 defacers accounted for the peak on 24 February, it was 22 on 26 February. As to defacements on Ukraine, no notable change was seen on the invasion day, but a peak of 69 attacks occurred 2 days later (6.30% of all defacements on that day, while it was 0.47% the day before). The largest wave was at around 7PM on 26 February (50 attacks), followed by medium waves at 5PM on 27 February (26 attacks) and 10PM on 3 March (29 attacks). The peak of defacers targeting Ukraine was on 27 February (1 day after the largest wave).

There was a spike of 771 defacements by 5 defacers targeting Russia on 25 May. Of these, 764 were claimed by a single defacer compromising a server hosting 760 sites. This outlier appears to be unique; it was removed from the graph for better visualisation.

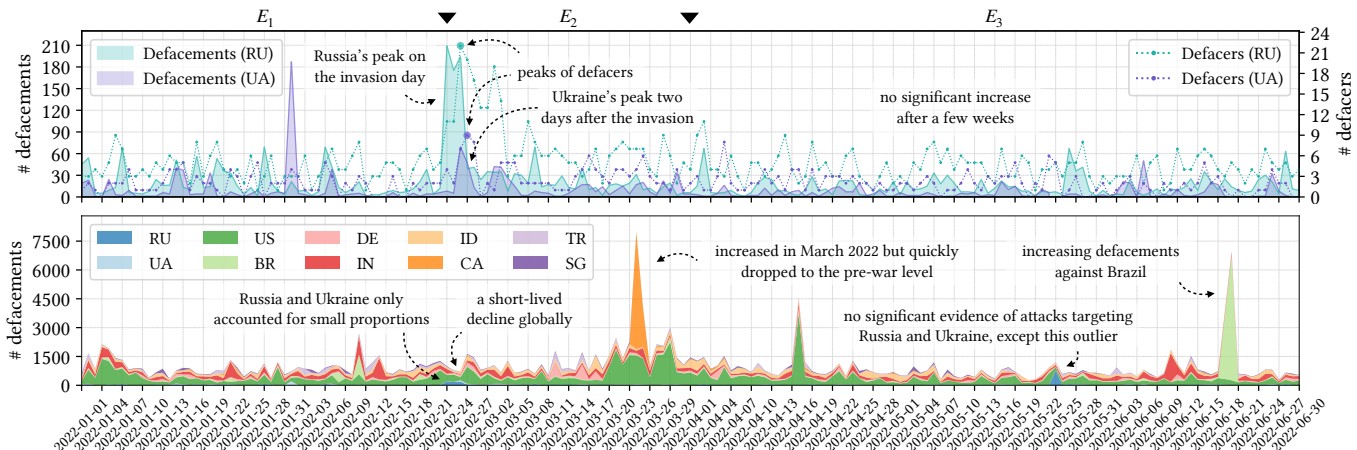

**Figure 1: Number of defacements and defacers per day in the Russia-Ukraine scale (top) and the global scale (stacked, bottom).**

**Table 2: Significance levels of the impact on daily defacements and defacers targeting Russia, Ukraine, and top countries.**

| Country | Tests for the number of web defacements per day | | | | | Tests for the number of web defacers per day | | | | |
|---|---|---|---|---|---|---|---|---|---|---|
| | ANOVA or Kruskal-Wallis | $\langle E_1, E_2 \rangle$ | $\langle E_1, E_3 \rangle$ | $\langle E_2, E_3 \rangle$ | $\eta^2$ | ANOVA or Kruskal-Wallis | $\langle E_1, E_2 \rangle$ | $\langle E_1, E_3 \rangle$ | $\langle E_2, E_3 \rangle$ | $\eta^2$ |
| Russia | $H(2) = 12.24, p < .01$ | $p < .05$ | $p = .3544$ | $p < .001$ | 0.06 | $H(2) = 26.57, p < .0001$ | $p < .0001$ | $p = .9083$ | $p < .0001$ | 0.14 |
| Ukraine | $H(2) = 17.86, p < .001$ | $p < .001$ | $p = .8377$ | $p < .0001$ | 0.09 | $H(2) = 13.64, p < .01$ | $p < .01$ | $p = .9286$ | $p < .001$ | 0.07 |
| US | $H(2) = 17.84, p < .001$ | $p < .01$ | $p = .1435$ | $p < .0001$ | 0.09 | $H(2) = 24.30, p < .0001$ | $p < .001$ | $p = .5961$ | $p < .0001$ | 0.13 |
| Brazil | $H(2) = 3.60, p = .1656$ | $p = .6481$ | $p = .1725$ | $p = .0912$ | 0.01 | $H(2) = 11.68, p < .01$ | $p = .3405$ | $p < .05$ | $p < .01$ | 0.05 |
| Germany | $H(2) = 3.43, p = .1796$ | $p = .2339$ | $p = .5269$ | $p = .0639$ | 0.01 | $F(2, 178) = 3.24, p < .05$ | $p = .7584$ | $p = .1858$ | $p = .0568$ | 0.04 |
| India | $H(2) = 4.21, p = .1221$ | $p = .9049$ | $p = .0670$ | $p = .1423$ | 0.01 | $H(2) = 3.90, p = .1424$ | $p = .0746$ | $p = .8734$ | $p = .0704$ | 0.01 |
| Indonesia | $H(2) = 10.90, p < .01$ | $p < .01$ | $p = .1566$ | $p < .05$ | 0.05 | $H(2) = 17.93, p < .001$ | $p < .001$ | $p = .5517$ | $p < .001$ | 0.09 |
| Canada | $H(2) = 8.13, p < .05$ | $p = .0944$ | $p = .2458$ | $p < .01$ | 0.03 | $H(2) = 9.51, p < .01$ | $p = .1020$ | $p = .1492$ | $p < .01$ | 0.04 |
| Turkey | $H(2) = 20.07, p < .0001$ | $p = .1171$ | $p < .0001$ | $p < .05$ | 0.10 | $H(2) = 13.26, p < .01$ | $p = .5501$ | $p < .001$ | $p < .05$ | 0.06 |
| Singapore | $H(2) = 3.83, p = .1473$ | $p = .7583$ | $p = .0677$ | $p = .2085$ | 0.01 | $H(2) = 5.90, p = .0524$ | $p = .3056$ | $p < .05$ | $p = .3218$ | 0.02 |

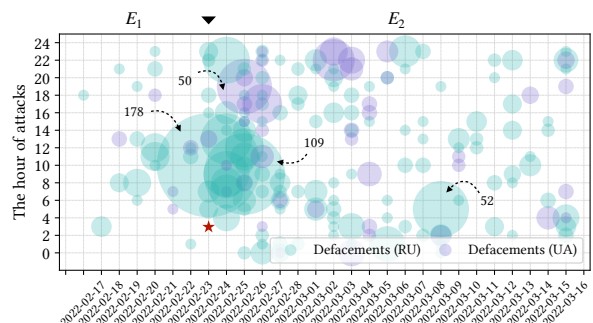

**Figure 2: Number of defacements hitting Russia and Ukraine by hour around the invasion day (marked with the red star).**

The peak of 187 defacements hitting Ukraine on 1 February 2022 by 4 defacers did not have a single cause and did not lead to a sharp increase in defacers in the following days.

For both Russia and Ukraine, Kruskal-Wallis tests suggest a statistically significant difference in the number of defacements and defacers per day through $E_1$, $E_2$, and $E_3$, see Table 2. Post-hoc analysis indicates a significant difference between the pre-invasion ($E_1$)

and one-month-post-invasion ($E_2$) periods. $p\langle E_2, E_3 \rangle$ is also significant, but *not* $p\langle E_1, E_3 \rangle$, suggesting the situation returned back to the pre-war levels after the second era. The effect sizes $\eta^2$ are all between medium and large, ranging [0.06, 0.14].

**The Global Scale.** The number of defacements against Russia and Ukraine are trivial when set against the global scale. Among 274 963 analysed defacements, only 5 899 (2.15%) targeted the two countries (4 340 for Russia and 1 559 for Ukraine). The top 10 countries account for 69.85% of all defacements, with sites hosted in the US consistently suffering the majority of defacements. Since January 2022, the US accounts for 26.95% of defacements, followed by India (11.47%) and Indonesia (8.41%), while Russia and Ukraine only account for 1.58% and 0.57%, respectively.

There was a short-lived decline in defacement attacks worldwide on the invasion day (from around 1 400 to 1 000), while it peaked for Russia from nearly zero to 209 (14.48% of all defacements). This suggests a genuine change in the way defacers chose their targets, precipitated by the war. The US is consistently the largest target, but only accounts for 21.97% on that day. During the last two weeks of March 2022, the number of defacements significantly increased at the global scale, with many defacements targeting the US. However, much like the Russia-Ukraine scale, the effect lasted for only a few

weeks. The unusual peaks against Brazil happened in late June (also for DDoS attacks, see Section §5), without a clear explanation.

The phenomenon seen from the Kruskal-Wallis and post-hoc tests in the Russia-Ukraine scale does not apply for most top countries, see Table 2. ANOVA/Kruskal-Wallis tests on the number of defacements and defacers are not all significant for Brazil, Germany, India, and Singapore; no significant changes are seen between the pre-invasion and one-month-post-invasion eras $\langle E_1, E_2 \rangle$ for Canada and Turkey. Indonesia has the similar phenomenon of post-hoc tests, yet one of the effect sizes is small. The only country following a close phenomenon is the US with medium effect sizes, yet Section §5 will point out this did not hold for DDoS attacks hitting the US.

The evidence above suggests a genuine increase of defacements against the two countries shortly after the invasion, significantly standing out from other top countries. Russia was the first to be hit at scale, followed by Ukraine a few days later. However, this effect was fairly short-lived for both countries, lasting for only a few weeks before returning to the pre-war levels, presumably as defacers ran out of targets or had lost interest in carrying out attacks. The number of involved defacers was small, but momentarily they turned from indiscriminate to more targeted attacks.

**Defacement Motives.** The conflict caught the attention of existing defacers, but also drew in new ones. While some minor players at the global scale made a significant contribution to the rise in attacks on Russia and Ukraine, the three most active defacers globally made a trivial contribution (less than 10) against either country (see Appendix §H). We do not verify findings on their general motives (see §2), but to gain conflict-related insights we analyse the contents of 4 340 defacements targeting Russia and 1 559 hitting Ukraine.

We annotate motives based on 1 341 unique messages left on the defaced pages. We consider a political sentiment and mark it as supporting Russia/Ukraine if a support/objection is expressed e.g., '*We stand with Ukraine!*'. We mark messages consisting of defacers' signature e.g., '*Hacked by Hero*' without clear motives, or just greetings to peers as being for fun or reputation. Messages advertising hacking tools and services or asking for ransom are marked financially motivated e.g., '*Contact me for shells*'. We label messages expressing favourite mottos or moods as self-expression e.g., '*Not much I want, hope my life will be better*', and exclude 1 278 messages (21.66%) containing empty or random messages.

We find diverse motives, but despite targeting Russia and Ukraine, most messages do not refer to the conflict. 2 723 (46.16%) were for fun/reputation, 1 219 (20.66%) self-expression, 143 (2.42%) related to other conflicts (such as Israel-Palestine), 58 (0.98%) related to patriotism, and 89 (1.51%) were financially motivated (mainly from the two most active defacers globally). Some defacers did leave conflict-related messages: 286 (4.85%) supporting Ukraine, roughly 2.8 times higher than those supporting Russia at 103 (1.75%). Notably, some defacers support Russia, yet also defaced Russian sites, saying they wished to alert and help secure the systems (22 attacks) – '*I have secured this domain, I love Russia*', was a message the third most active pro-Russia defacer left on a Russian website. Likewise, other defacers supported Ukraine yet defaced Ukrainian sites (12 attacks) e.g., '*Hello Volodymyr Zelensky, I'm sorry to hack your site. I just wanted to tell you that people need a president like you. We support Ukraine*'. Such signatures are likely intentionally war-related as Russia and Ukraine were not frequently targeted before.

## 5 THE EVIDENCE FROM DDOS ATTACKS

We now examine if there were also significant changes in DDoS attack volumes targeting Russia and Ukraine after the conflict. Figure 3 shows the number of DDoS attacks in both Russia-Ukraine and global scales over the three eras, while Figure 4 shows their changes by hour during the most active four weeks from 24 February.

**The Russia-Ukraine Scale.** DDoS attacks lagged defacement by about a week, but occurred in higher volumes and lasted longer; most happened after 7AM. The number of both DDoS attacks and victims targeting Russia first increased on 2 March (6 days after the invasion) with 851 victims, 511 of them at around 6PM. The attacks peaked 4 days after with 1 137 victims. High activity levels continued through 23 March, with the biggest wave occurring at around 2PM on 8 March with 755 victims. Smaller waves continued regularly during the next few weeks. Regarding DDoS attacks hitting Ukraine, significant waves started around a week after Russia's first big wave (some small spikes targeting Ukraine before Russia were insignificant) with the first notable spike on 10 March having more than 526 victims, then became prevalent during two weeks from 18 to 31 March: big waves were on 18 March at around 12PM, 1PM and 4PM with 257, 476, and 700 victims, respectively. Other big and medium waves lasted until the end of March, with the biggest peak on 31 March when 1 296 victims were hit. The increased volume only continued for about a month before declining sharply.

Kruskal-Wallis tests suggest statistically significant changes between the daily number of DDoS attacks and victims through the three eras for both Russia and Ukraine, much like what we see with defacements, see Table 3. Post-hoc analysis shows high significance levels of $\langle E_1, E_2 \rangle$ and $\langle E_2, E_3 \rangle$, suggesting notable changes between the pre-invasion vs. one-month-post-invasion periods, and the one-month-post-invasion periods vs. the period after that. The main difference between Russia and Ukraine here is that the situation for Ukraine returned to pre-invasion levels after one month i.e. $\langle E_1, E_3 \rangle$ is not significantly different, while we still see some difference with Russia. Indeed, the number of DDoS attacks hitting Russia was still slightly higher than before the invasion (see Figure 3). The effect size is large for Russia, while it is medium for Ukraine.

**The Global Scale.** We again see concentrations in DDoS attacks, with the top 10 countries accounting for 70.49% of all victims. The US still dominates (24.68%), followed by Brazil (11.99%) and Bangladesh (8.10%). Ukraine took 1.57%, while Russia lies 8th at 3.61%. Our DDoS and defacements datasets show some correlations. Three of the top 10 countries for defacements are also in top 10 for DDoS targets: the US, Germany, and Brazil. As with defacements, the number of DDoS attacks rose globally during the last 2 weeks of March 2022. The volume hitting Bangladesh is insignificant, see Table 3. The unusual peaks of both defacements and DDoS against Brazil in late June are notable; Brazil is often ranked among top cybercrime hubs worldwide [40], yet we lack a convincing causality. A similar peak observed in the Russia-Ukraine scale can also be seen at the global scale following the invasion. Similar to defacements, DDoS attacks thrived on a global scale in March, yet they quickly returned to their previous levels after a few weeks.

The phenomenon seen from the Kruskal-Wallis and post-hoc tests in the Russia-Ukraine scale does not apply for most top countries, see Table 3. Kruskal-Wallis tests are not *all* significant for

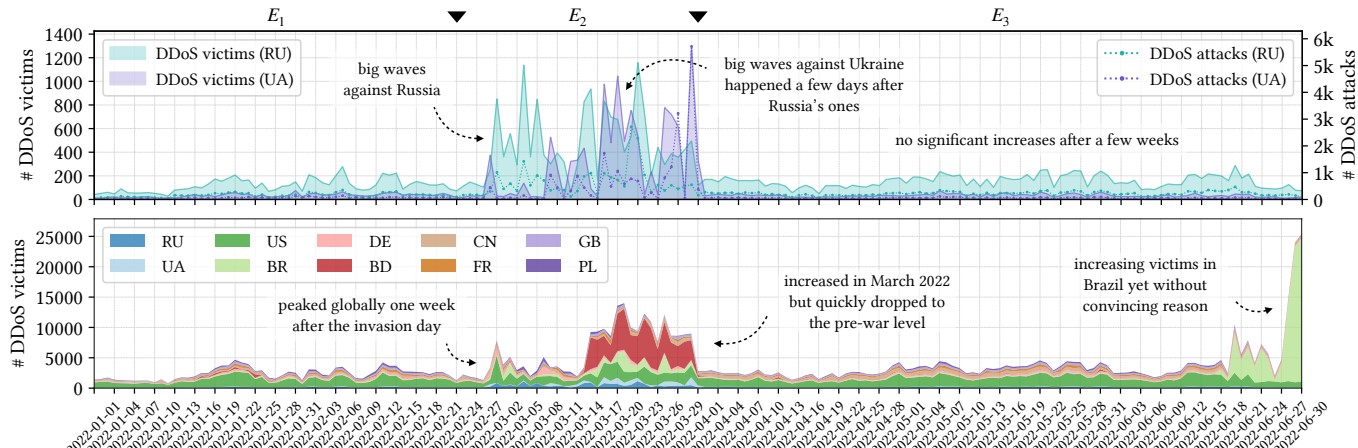

Figure 3: Number of DDoS attacks and victims per day in the Russia-Ukraine scale (top) and global scale (stacked, bottom).

Table 3: Significance levels of the impact on daily of DDoS attacks and victims targeting Russia, Ukraine, and top countries.

| Country | Tests for the number of DDoS attacks per day | | | | | Tests for the number of DDoS victims per day | | | | |
|---|---|---|---|---|---|---|---|---|---|---|
| | ANOVA or Kruskal-Wallis | $\langle E_1, E_2 \rangle$ | $\langle E_1, E_3 \rangle$ | $\langle E_2, E_3 \rangle$ | $\eta^2$ | ANOVA or Kruskal-Wallis | $\langle E_1, E_2 \rangle$ | $\langle E_1, E_3 \rangle$ | $\langle E_2, E_3 \rangle$ | $\eta^2$ |
| Russia | $H(2) = 60.67, p < .0001$ | $p < .0001$ | $p < .0001$ | $p < .0001$ | 0.33 | $H(2) = 57.13, p < .0001$ | $p < .0001$ | $p < .01$ | $p < .0001$ | 0.31 |
| Ukraine | $H(2) = 12.59, p < .01$ | $p < .01$ | $p = .4593$ | $p < .001$ | 0.06 | $H(2) = 15.16, p < .001$ | $p < .001$ | $p = .8765$ | $p < .001$ | 0.07 |
| US | $H(2) = 6.98, p < .05$ | $p = .5592$ | $p < .05$ | $p = .1182$ | 0.03 | $H(2) = 4.43, p = .1093$ | $p = .2527$ | $p < .05$ | $p = .5594$ | 0.01 |
| Brazil | $H(2) = 9.81, p < .01$ | $p < .01$ | $p = .2006$ | $p < .05$ | 0.04 | $H(2) = 13.12, p < .01$ | $p < .01$ | $p < .05$ | $p < .05$ | 0.06 |
| Germany | $H(2) = 9.49, p < .01$ | $p < .01$ | $p < .01$ | $p = .6609$ | 0.04 | $H(2) = 17.24, p < .001$ | $p < .001$ | $p < .001$ | $p = .2039$ | 0.09 |
| Bangladesh | $H(2) = 3.96, p = .1379$ | $p < .05$ | $p = .2198$ | $p = .2785$ | 0.01 | $H(2) = 4.43, p = .1090$ | $p < .05$ | $p = .1353$ | $p = .3585$ | 0.01 |
| China | $H(2) = 80.16, p < .0001$ | $p < .001$ | $p < .0001$ | $p < .0001$ | 0.44 | $H(2) = 65.91, p < .0001$ | $p < .0001$ | $p < .0001$ | $p = .0674$ | 0.36 |
| France | $H(2) = 16.04, p < .001$ | $p = .2586$ | $p < .001$ | $p < .05$ | 0.08 | $H(2) = 9.96, p < .01$ | $p = .1519$ | $p < .01$ | $p = .2366$ | 0.04 |
| UK | $H(2) = 13.90, p < .01$ | $p = .4892$ | $p < .001$ | $p < .05$ | 0.07 | $H(2) = 7.94, p < .05$ | $p = .5258$ | $p < .01$ | $p = .0976$ | 0.03 |
| Poland | $H(2) = 0.34, p = .8423$ | $p = .8841$ | $p = .6759$ | $p = .6002$ | 0.00 | $H(2) = 0.04, p = .9809$ | $p = .9081$ | $p = .9365$ | $p = .8449$ | 0.00 |

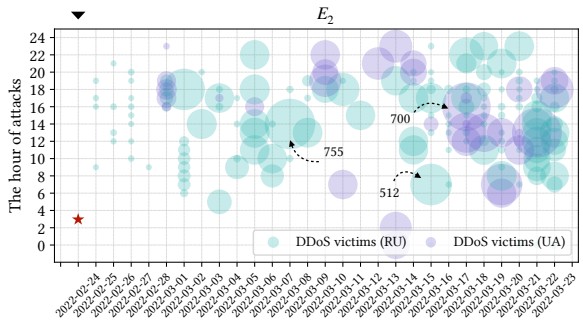

Figure 4: Number of DDoS victims in Russia and Ukraine by hour around the invasion day (marked with the red star).

the US, Bangladesh, and Poland, despite the US accounting for the largest number of attacks and there was a visual increase for Bangladesh (as the tests compare mean ranks instead of means). No significant changes are seen between the pre-invasion and one-month-post-invasion periods $\langle E_1, E_2 \rangle$ for France and the UK. Brazil and Germany have a similar phenomenon of post-hoc tests in the Russia-Ukraine scale, yet one effect size is small. China is the only

country following that phenomenon with large effect sizes; the main difference is that the changes in $p\langle E_2, E_3 \rangle$ is not significant.

Much like defacements, the evidence above suggests a genuine increase of DDoS attacks targeting Russia and Ukraine as the conflict began, significantly standing out from most top countries. Russia was still the first to be hit at scale, followed by Ukraine shortly after. The main difference with defacements is that DDoS attacks hitting Russia were not entirely back to the previous levels, but were slightly higher. The outbreak of both defacement and DDoS attacks against Russia and Ukraine was significant and timely, but fairly short-lived: it returned to pre-war levels after just a few weeks.

## 6 THE HACKING COMMUNITY REACTIONS

**Discussions on Hack Forums.** There was an immediate increase of Hack Forums posts mentioning the two countries after the invasion, from near zero to over 120 per day, see Figure 5. Kruskal-Wallis tests confirm the significance $H(2) = 72.98, p < .0001$, with a large effect size $\eta^2 = 0.40$; pairwise post-hoc tests for $\langle E_1, E_2 \rangle$ and $\langle E_2, E_3 \rangle$ are both significant ($p < .0001$), but not $\langle E_1, E_3 \rangle$ ($p = .8501$). The number of posting users shows a similar story: Kruskal-Wallis test reports $H(2) = 77.54, p < .0001$ with a large effect size $\eta^2 = 0.42$; pairwise post-hoc tests for $\langle E_1, E_2 \rangle$ and $\langle E_2, E_3 \rangle$ are both significant

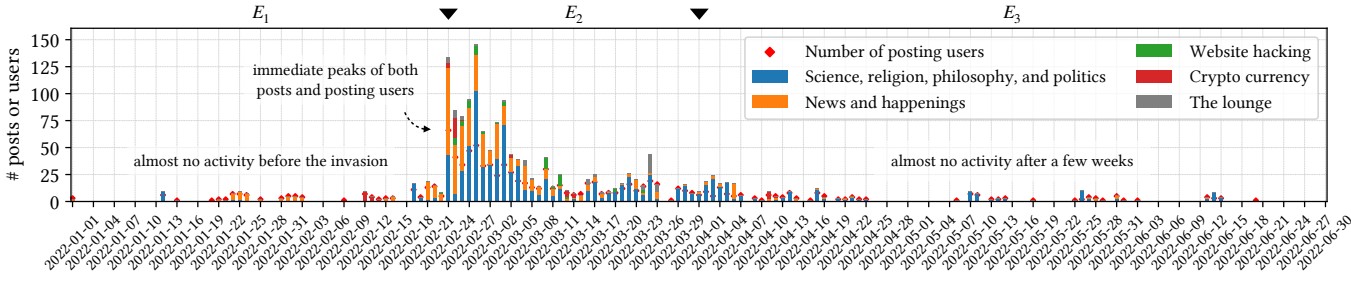

**Figure 5: Number of daily posts and posting users on Hack Forums mentioning Russia and/or Ukraine (top five subforums).**

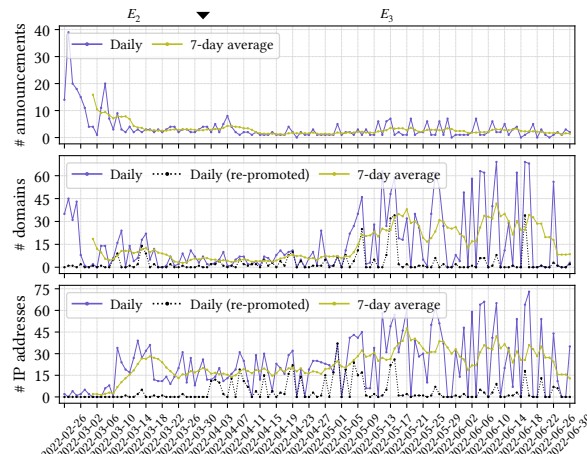

**Figure 6: The number of announcements and (re-promoted) targets in the IT Army of Ukraine Telegram channel by day.**

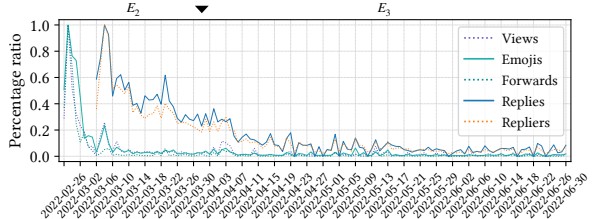

**Figure 7: Levels of user engagement daily in the IT Army of Ukraine Telegram channel. Values are min–max normalised.**

($p < .0001$), but not $\langle E_1, E_3 \rangle$ ($p = .6657$). This fits the evidence seen with defacement and DDoS attacks: both posting activity and users returned to the pre-war level after a few weeks, presumably as users lost their interest and moved on to other discussion topics.

This posting volume is tiny when set against the 62M-post size of HACK FORUMS, showing trivial contributions of the Russia-Ukraine discussions in the overall landscape (similar to the previous evidence seen from defacement and DDoS attacks). These posts are centralised: 97.22% belongs to the top 5 popular subforums. Ranked 1st is *'science, religion, philosophy, and politics'*, accounting for 53.40%; ranked 2nd is *'news and happenings'* with 33.28%; *'website hacking'* ranked 3rd, followed by *'crypto currency'*, then general chats. We see some *'news and happenings'* posts in the past, but mostly no *'science, religion, philosophy, and politics'* posts until the invasion.
**Targets Promoted by the IT Army of Ukraine.** Many announcements and targeted domains are posted in the first 2 weeks after the invasion, beginning on 26 February, peaking on 27 February with 40 announcements and 45 domains promoted (IP addresses were not regularly included until later), see Figure 6. Yet, they quickly declined to consistently less than 10 per day after two weeks with some days (e.g., 24 and 26 April) having no posts. The number of subscribers also dropped from 300k to around 160k in October 2023.

While the number of announcements dropped, the number of targets has steadily increased, particularly in May and June 2022

with multiple-target posting. Activities were unstable at that time; targets got promoted less frequently and occasional days had no targets. Targets were mostly fresh in the first 2 weeks, but then a considerable proportion got re-promoted on multiple days e.g., all advertised IP addresses and most domains were re-posted during 4–6 May. Along with frequent zero-target days, this suggests the group might run out of new targets or get bored with finding them.

Community reactions and engagement tell much the same story as with DDoS and defacement attacks (see Figure 7). While more targets were promoted in May and June, volunteers appeared to have largely lost interest, despite their intense activity in the first few weeks. The decline in reaction was consistent across all engagement types: views, emojis, forwards, and replies. Older announcements may have more time to accrue views as people scroll up the channel, but the emojis, forwards, and replies require user intent. We believe the figures reflect a genuine decline in engagement over time.

We further looked at user engagement with instructions about tools and guidance to carry out attacks provided by the group. The group first provided material to hit Russian payment system on 9 March (2 weeks after the invasion), attracting high levels of engagement: 240k views, 2.6k emojis, 1.2k forwards, and 421 replies from 197 users. The next was on 1 April: while the number of replies and forwards was similar to the first, other kinds roughly halved. From mid-May to late June, instructions were posted 4 more times, yet users were around 4 times less engaged than the first in March, indicating a loss of interest despite the operator's extensive efforts.
**Target Selection.** Targets were often themed, sometimes patterned around particular weekdays e.g., online news and propaganda, food delivery services, entertainment are often hit at weekends to maximise impact as people spend more time online. Themes were also occasionally set with re-promoted old targets, leading to wide variations in the number of new targets, particularly from May onwards

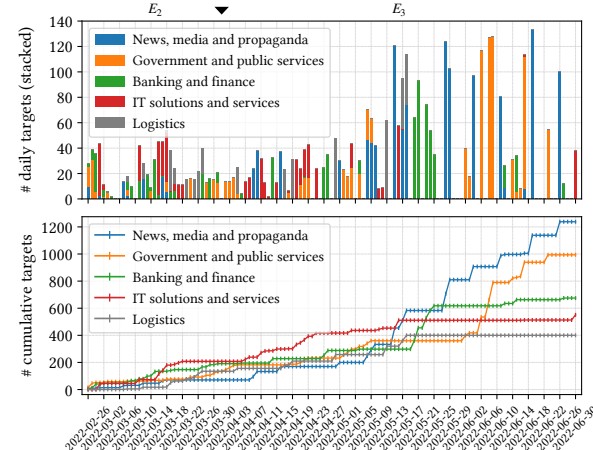

**Figure 8: Number of daily stacked targets (top) and cumulative targets (bottom) being promoted in top five categories.**

(zero on some days), see Figure 6. Subscribers can suggest new targets, but the group owner posts most of them; during the last 2 months, they often re-promoted targets linked to old posts, which could be as simple as '*we continue to work with yesterday's targets*'.

We use categories linked with targets by default; when unavailable, we rely on root domains e.g., .tv and .gov are likely news and government sites. Categories of generic domains (e.g., .net, .com) are identified by direct visits (via Russian IP relays) or querying Internet archives if they are down. Some targets were indeed down while previously active, suggesting attacks might have succeeded e.g., ksrf.ru (the Constitutional Court of the Russian Federation) was down for a while, and data.gov.ru was both defaced and DDoSed.

Categories vary, yet five dominate 80.21% of all targets, see Figure 8. '*News, media and propaganda*', including TV broadcasting, has been consistently promoted since the war began, but only became the most common one in May when it overtook '*IT solutions and services*'. '*Government and public services*', which includes military, state-owned websites, and public services for civilians such as parking and lighting (including governments imposed on occupied territories) has also been regularly targeted throughout, but they only grew rapidly towards the end of the period, making it the second most common category overall. '*Banking and finance*' ranked 3$^{rd}$, including banks, stock exchanges, electronic payment, accounting, credit services, trading, bidding, investment platforms and funding agencies. '*IT solutions and services*' ranked 4$^{th}$, including software solutions supporting governments, digital signature and information security services such as DDoS-Guard. It was actively promoted early on but was targeted far less thereafter. '*Logistics*' ranks 5$^{th}$, including airlines and aviation, travel, shipping, and food delivery. Other popular categories include markets and stores (e.g., job markets, real estate, e-commerce, drug stores), manufacturers and trading (e.g., military footwear, wood and roofing materials), education, insurance, telecoms (e.g., Internet providers), businesses and state companies (e.g., energy and steel manufacturers), forums, entertainment (e.g., cinemas), and non-governmental organisations. **Crossover with Observed Attacks.** The IT Army of Ukraine maintains a dashboard of targets' status, claiming many are down due to

their actions. To find whether the attacks involved reflected DDoS or defacement, we correlate our attacks records with promoted targets since the Telegram group started. We consider a defacement overlap when either its URL or IP address matches promoted targets, while for DDoS attacks, only IP addresses are used.

There was very little overlap with defacement: among 3 845 promoted targets, there are only 59 valid matches (1.53%), including 7 domain matches (0.18%) and 52 IP matches (1.35%). Notably, no overlaps occur on the day targets are promoted, suggesting that defacers chose their targets themselves independently; these targets are largely unimportant and irrelevant to the conflict. For DDoS attacks, we observe 707 (30.86%) total overlaps among 2 291 promoted IP addresses, which is considerable. Unlike defacements, some are executed the same day they are promoted; we find many same-day overlaps in late March, early April and during May, peaking on 19 March 2022 with 22 victims overlapping. However, the crossover dropped quickly, becoming less frequent from late May while many new targets were still actively advertised. This suggests a loss of interest by volunteers in attacking targets promoted by the group.

## 7 CONCLUDING REMARKS

The role of the low-level cybercrime actors studied in this paper – which we believe are meaningfully measurable – are presumably trivial acts of solidarity and opportunistic competition. We found little measurable evidence to suggest these actors are making any persistent contribution to the conflict, even in a major war between two nations with a long history of cyberwarfare. Their role and capacity in nation-state conflicts that might be seen in the future should not be collapsed together with state hacking and political 'hacktivism'. Our diverse, separately collected datasets all point to a narrative that notable attention was temporarily drawn to Russia and Ukraine but not other countries. Neither the engagement on Hack Forums nor Telegram, the outbreak of defacements nor DDoS attacks was long-lasting, presumably as participants simply lost interest, despite their choice of targets being clearly influenced by the war for a moment. This is in line with other work suggesting that boredom is an important factor in people leaving cybercrime [11].

We do not dispute claims about the prevalence of state-sponsored attacks such as malware and phishing [21, 43], but rather provide additional perspectives on the role of many low-level actors. Some cybercrime-related activities are indeed contributing to the war effort. Leaks, especially of high-profile datasets gathered from Russian public services, have consistently made headlines. They may or may not be connected to civilians, hacktivists, state actors, or other groups. Much as with ransomware, their low numbers and vast disparities in impact make them far less cross-comparable. Our findings fit a more general pattern in the cybercrime ecosystem increasingly characterised by an entrepreneurial, service-based economy which is becoming alienated from traditional hacker culture's concerns with technical learning and dissent [3]. Committed, persistent hacktivists appear to be separate from the low-level crime communities whose interest seems to have been fleeting and easily diverted by trending news. They were indeed briefly getting involved in the war effort by using off-the-shelf exploit tools, but their role on the 'hard' digital frontline remains limited – these are likely actions in the theatre of protest, 'soft power' and solidarity.

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

# A ETHICAL CONSIDERATIONS

This work is presented objectively to minimise risk to researchers. The collection, sharing, and analysis of web defacement, amplified DDoS attacks, and Telegram chats have been formally approved by our ethics committee. We do not attempt to gather private data; only publicly accessible data are collected. A 2022 US court ruling found scraping public data is legal [68]. Our scraper does not overload websites. The amplified DDoS attack honeypots absorb attack packets without relaying them, thus reducing harm to victims.

Studying an ongoing conflict may harm the individuals whose attacks are reported and the researchers who might face retaliation from attackers due to leaking insights into their activities and community. To avoid those potential harms, we carefully designed our experiments to operate ethically and collectively without any findings linked to individuals. We did not ask for consent from Telegram users or web defacers when using scraped data, as sending thousands of messages would be impractical. We assume they are aware that their messages are publicly visible after publishing. This approach accords with the British Society of Criminology's Statement on Ethics [9], as all analysis involved aggregated data, without identifying individuals.

# B DATA LICENSING

We have robust procedures and long experience in making our data available in various jurisdictions. Our quantitative data and analysing code are available for academic researchers under a license agreement with *Anonymised Centre* to prevent misuse and to ensure the data will be treated ethically, as accessing sensitive data might risk both researchers and the involved actors [75].

# C DETERMINING ATTACK GEOLOCATION

Accurately mapping IP addresses to countries is challenging, as IP geolocation is not always stable and trustworthy [22]; providers prefer locating servers in countries with cheap hosting [76]. Attack geolocation might thus be determined differently by different archives e.g., Zone-H may say an IP is in Germany, while Zone-Xsec goes for Singapore and Defacer-ID cannot tell. Geolocation services are more reliable at the country level [14], but this is only part of the truth as websites are nowadays commonly hosted on content delivery networks (CDNs). The original IP addresses are typically hidden and the geolocation is of the CDNs.

For example, a '.ru' website is supposed to be Russian, but it might be physically hosted in Vietnam, operated by a person living in Hong Kong, while proxied through Cloudflare with an IP address in the US. Relying on only one aspect might be risky, as both IP and domain can lie. We use data fusion to enhance the accuracy, prioritising: (1) top-level domain; (2) IP geolocation at collection time (MaxMind GeoIP2[1] for web defacement, and a database we maintain based on Regional Internet Registry data for DDoS attacks; (3) geolocation of the AS hosting the IP address. If a website's IP address belongs to a CDN, its geolocation is determined solely by ccTLD, as any geolocation of IP address or ASN will be unreliable.

The top three CDNs are Cloudflare, Amazon Web Services, and Akamai, serving around 89% of customers [29]. We ignore the trivial market shares of their competitors, but we count DDoS Guard as it is based in Russia, which may affect the infrastructure hosted there. We expect the four can cover nearly 90% of customers. In total, we found 4.87% of defacements are hosed on these CDNs by 14 262 prefixes as of the writing date: 1 698 of Cloudflare; 7 483 of Amazon Webservice; 5 056 of Akamai; and 25 of DDoS-Guard (these prefixes and AS number mappings are collected on Hurricane Electric Internet Services). For defacements, we prefer ccTLD over IP geolocation as attackers likely target websites in a country by massively scanning domain ccTLD (e.g., '.ru', '.ua') rather than checking if IP addresses are hosted in that country.

Accurate measurement of frequent ccTLDs used in Russia and Ukraine is complex; many Ukrainian firms use Russian services and vice versa. The most frequented domain used in Russia is reported '.ru' [66]. We cannot find a similar report for Ukraine, yet we believe incorporating ccTLDs with IP and AS geolocation is fairly reasonable as choosing targets based on ccTLDs is a straightforward way used by low-lever cybercrime actors.

## D  DEFACEMENT SUBMISSION PROCESS

The defacement submission is mostly automatic: users specify a 'notifier', team information, defaced URL, vulnerability types, and hacking incentives. New 'on hold' reports are kept away from the dashboard until being verified by staff or bots. At that point, a record is made with details of the compromised system, its IP address and location, and a snapshot of the defaced page (often consisting of the defacer's messages, which may include political and ideological propaganda [6]). Although 'notifier' can be arbitrarily entered, defacers are incentivised to use a consistent handle to cultivate fame and reputation. We thus consider 'notifier' to be reliable enough to differentiate between defacers. Snapshots of defaced sites, including messages left are highly reliable, as they are captured at reporting time. Messages can be hidden by using identical font colours as the background, but are detectable by analysing the HTML.

## E  WEB DEFACEMENTS COLLECTION

Data completeness and reliability are critical for longitudinally measurement. Scraping *complete* snapshots, especially Zone-H, is non-trivial, and was not guaranteed in prior work. Some attempted to purchase Zone-H snapshots [41], but this is not sustainable, and

---

[1] GeoIP2 is freely accessible at https://maxmind.com/. It offers both free and paid licenses, with the paid one being slightly more accurate and up-to-date. It claims to provide over 99.8% country-level and over 60% city-level accuracy, yet that varies from country to country e.g., 79% for Russia and 65% for Ukraine, within a 250km radius.

---

**Algorithm 1** Semi-automatic defacement validation

1: **procedure** VALIDATE_DEFACEMENTS
2:    **for each** $a \in$ verifiedDefacements() **do**  ▷ verified by archives
3:        a.status $\leftarrow$ 0  ▷ originally validated
4:    **end for**
5:    **for each** $a \in$ filteredDefacements() **do**  ▷ filtered by terms
6:        a.status $\leftarrow$ 1  ▷ automatically validated
7:    **end for**
8:    $P \leftarrow$ pendingGroups()  ▷ groups of pending attacks
9:    $V \leftarrow$ verifiedGroups()  ▷ groups of verified attacks
10:    **for each** $p \in \mathcal{P}$ **do**
11:        $T \leftarrow \{\}$
12:        **for each** $v \in \mathcal{V}$ **do**
13:            $d \leftarrow$ levenshtein($p, v$)  ▷ Similarity with verified ones
14:            $T \leftarrow$ topSimilar($d, T$)  ▷ Extract top similar ones
15:        **end for**
16:        showSimilarDefacements($T$)  ▷ to assist the annotators
17:        $s \leftarrow$ annotation()  ▷ annotate the validity
18:        **for each** $a \in p$ **do**
19:            a.status $\leftarrow s$  ▷ update validation status
20:        **end for**
21:        **if** isValidated($s$) **then**  ▷ if it is manually validated
22:            $V \leftarrow V \cap p$  ▷ add to validated groups
23:        **end if**
24:    **end for**
25: **end procedure**

---

is ethically questionable. Gathering defacement archives at scale is challenging as (1) Zone-H adopts text Captcha to prevent bots, (2) its dashboard sets a limit of 50 pages where older data is hidden, and (3) on-hold records may not appear promptly, leading to potential misses. The only way to get a complete scrape is by iterating through all submission IDs (this generated non-trivial workload) with the IDs of valid and invalid reports often mixed. Dealing with these issues, plus IP blacklisting and bot prevention mechanisms, is the main challenge to scraping. We responded by (1) developing an efficient text Captcha solver for Zone-H utilising image-processing techniques, (2) routing our scraper through multiple proxies, and (3) carefully iterating through all submission IDs in turn. We stored raw data in a database to avoid unnecessary requests in the future.

Five most trusted archives are included; an active one DEFACER-ID (since February 2016) is excluded as (1) the valid submission volume during the period is small (less than 27K); (2) unclear staff verification, no validity sanitisation on submission, no validity signal in defaced pages (in fact, over half of these have been deemed invalid by the archive); (3) defaced snapshots and defacers' messages are missing; and (4) victim geolocation is mostly lacking; determining it after the fact is problematic as sites could have been relocated.

## F  VALIDATING ON-HOLD DEFACEMENTS

How defacement submissions are validated is not clearly stated. While Zone-H reports are kept on hold until being manually verified by staff, Zone-Xsec, Defacer-Pro, and Haxor-ID use automatic validation, insisting messages left on the defaced pages contain keywords linked to hacking activities (e.g., *'Hacked by Me'*). Defacers may game the system by putting comments on blogs, or submitting search queries (e.g., ?search='Hacked by Me'), which

occasionally get through automatic sanitisation but our further validation excludes them. Manual staff review on ZONE-H may be slow, while automatic verification of the others is error-prone. Unverified records may be kept on-hold forever, leading to incomplete data. Consequently, collecting only defacements shown in the dashboard is inadequate, making a complete dataset challenging to gather. To enhance data completeness and reliability, a semi-automatic validation is performed to check if on-hold reports are in fact valid.

Our our strategy is shown in Algorithm 1. First, reports verified by archives are considered valid. Second, messages on the defaced pages of on-hold submissions are used to decide the validity, as defacers often leave signatures for reputation e.g., *'Hacked by CoolHacker'*. If messages include defacers' handles and specifically contain common hacking terms: *'hacked by'*, *'h4ck3d by'*, *'h4cked by'*, *'p4wn3d by'*, *'pwn3d by'*, *'pwnd by'*, *'pwned by'*, *'pwndz by'*, *'owned by'*, *'own3d by'*, *'touched by'*, and *'kissed by'*, we consider them to be valid e.g., a message *'This website was hacked, contact me t.me/coolhacker'* posted by a notifier *'CoolHacker'* is considered valid. This method is looser than an exact comparison with *'Hacked by CoolHacker'*, but is still highly accurate; 100 randomly checked samples were all correct. Third, the remaining submissions are manually validated by looking for defacers' signature; some are obvious and some are complicated. Candidates are grouped by normalised handles and messages (redundant spaces removed), then for each, 10 most similar validated defacements are suggested to the annotator. Levenshtein distance is used to estimate the similarity between two messages, which is helpful as messages are often slightly modified from existing templates. If no message is found (instead images, iframes, or javascript), or leftover signatures cannot be spotted, a web browser opens the defaced page and assist annotators.

This assistance effectively reduces the annotator's effort. One challenge is the redirects to a defacer's page, as this can be modified dynamically; when the defacer's page is down, the submission points to a non-existent site, but a careful check could reveal evidence of the defacers. We also consider a site is touched if its content is unchanged but the page title is modified to indicate hacking activities. We ignore cases that lack evidence to ensure that those flagged 'valid' are indeed valid. We do not use complex machine learning techniques as message texts contain lots of noise; given a small number of samples (around 10k), machine learning is not more effective than a rule-based approach. Sometimes defacements appeared to be already verified at the time of collection, but became invalid afterwards; we re-validate them months after collection to make sure their status has been finalised by the archives.

## G  UNIFYING DEFACEMENTS AND DEFACERS

We hashed then unified defacements based on the reporting date, original defacer handles, root victim domain, and message left on the defaced page. Including reporting dates may be problematic if defacers resubmit to other archives after a few days; but excluding them may lead to spotting matched submissions on different days due to repeat victimisation. We also unified defacers across all archives, as users tend to pick similar pseudonyms on different platforms [20]. As the unification needs to be accurate and the number of unique defacers is just a few thousand, machine learning is not appropriate. We instead used a semi-automated approach

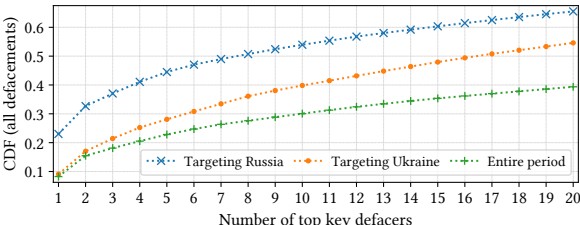

**Figure 9: The concentration of defacers in the entire period, and those targeting Russia and Ukraine after the war began.**

combining automated handle similarity analysis with manual review. First, similar pairs of handle are extracted using Levenshtein distance, which is set to not exceed 25% of the handles' length. Then, 10 messages left by defacers in each pair are sampled to assist the annotation; the pair is unified under a single nickname if their messages are semantically closed enough. The decision is based on message patterns, stylometry, synonyms, the handle inclusion in the messages, typos, team, nationality, messages' semantics, language, and the handle rarity (rare ones like 'cj2ks' are more likely to be used by a single person, while common ones like 'glory' are more likely to be shared by multiple individuals [39]). Many handles leaving similar messages across different archives, while many different typos occur e.g., missing characters, order of characters, case-sensitive. Messages left are diverse: some are identical, some are relatively similar, some are distinct and contain the defacers' name, and some come with phone numbers. We only confirm when having sufficient evidence, uncertain pairs are left unmatched.

## H  KEY DEFACERS

Key actors play central roles in underground communities; a small number of actors are often involved in many activities [27, 72]. Reflective DDoS attacks lack attacker identifiers, so can only investigate 'key defacers'. Figure 9 shows the number of key defacers and the proportion of defacements they contributed. We found a high concentration: over 6 months, 10 defacers accounted for 30.06% of attacks, while the most active of them contributed 8.25% (around 22.7k). If we ignore the pre-invasion period, conflict-related defacements show higher concentrations: the top 10 targeting Ukraine accounted for 39.82%, while the most active was responsible for 9.10%. The effect is even more pronounced for defacements targeting Russia, where the numbers are 53.95% and 23.01%, respectively.

Among the most active defacers in the entire 6 months, two actively attacked both Russia and Ukraine when the war began: the 5th ranked 3rd & 4th, and the 9th ranked 4th & 1st, for attacking Russia and Ukraine, respectively. Some picked sides: the 8th ranked 7th for attacking Russia, while the 4th ranked 5th for targeting Ukraine. The 6th did not target either country at all. We found some 'new faces' e.g., the second most active defacer targeting Russia after the war began first appeared in mid-February, peaked on the invasion day, stayed significant for 3 days then declined quickly. Some 'old faces' performed many attacks against other countries but not Russia and Ukraine until suddenly after the invasion, suggesting this influenced their choice of targets.

