# OpenReview forum: "Getting Bored of Cyberwar: Exploring the Role of Low-level Cybercrime Actors in the Russia-Ukraine Conflict"
_ACM.org/TheWebConf/2024/Conference — TheWebConf24 Oral_

### Official Review · Reviewer_kCTa · 2023-11-09

**Novelty:** 6
**Technical Quality:** 7

**Review:**

This paper delves into a crucial subject, examining the involvement of regular users and hacktivists in cybercrime amid conflicts, with a focus on the Ukraine-Russia conflict. Following this event, a surge in cyberattacks, predominantly targeting Russia, was observed. The authors conducted a thorough analysis by collecting and examining data from archives, forums, and Telegram group, specifically exploring attacks like defacement and DDoS incidents related to the Ukraine-Russian conflict. The study considered three distinct periods for analysis: one month prior to the invasion, around one month after the invasion, and two months post the invasion.

Although users engaged with the cause, the findings revealed a declining trend in user engagement with these attacks over time. The authors suggested a potential explanation for this decline: waning interest or boredom among individuals associated with the cause.

Comments to the authors: I found the paper to be highly engaging, with a straightforward narrative that facilitated comprehension. The authors executed exemplary data analysis, offering statistical significance to back their results. The paper is well-written and provides an interesting contribution to the in-depth exploration of cybercrime activities.

The identified drawback of this paper lies in its data constraints, a limitation inherent to the nature of the analysis, a point duly acknowledged and addressed by the authors. So I have no concerns about it.

**Questions:**

More like a comment instead of a question: Out of curiosity, if the authors intend to continue this work, perhaps it is worth analyzing the Telegram groups at scale. Drawing inspiration from prior WWW works on WhatsApp and Telegram groups, albeit focused on political content, could serve as a valuable methodological inspiration.

**Ethics Review Description:**

No issues

**Reviewer Confidence:**

3: The reviewer is confident but not certain that the evaluation is correct

**Scope:**

4: The work is relevant to the Web and to the track, and is of broad interest to the community

---

### Official Review · Reviewer_dm3F · 2023-11-11

**Novelty:** 4
**Technical Quality:** 6

**Review:**

The paper studies the cyberwar between Russia and Ukraine.
They study acts of cyberwar before and after the invasion.
The information comes from the following sources: web defacements, DDOS attacks, and discussions.

This is a very interesting paper.

Did you try to corelate the cyberwar events that you study with the intensity of the traditional war activities? Do they peak at the same time intervals?

The paper concludes that people get bored of cyberwar. It is not clear what is the evidence for this. Do you consider other alternatives? e.g. that their cyberwar acts did not make any difference?

There are other cyberwar activities (such as APTs) that may have a higher impact. These usually keep a low profile. Did you study any of these?

**Questions:**

See above.

**Ethics Review Description:**

The authors address ethics concerns in Appendix A.

**Reviewer Confidence:**

3: The reviewer is confident but not certain that the evaluation is correct

**Scope:**

4: The work is relevant to the Web and to the track, and is of broad interest to the community

---

### Official Review · Reviewer_d2LC · 2023-11-16

**Novelty:** 6
**Technical Quality:** 5

**Review:**

Thank you for submitting the manuscript to theWebCon’24!

The paper explores if and how the military conflict between Russia and Ukraine was expanded to the Web. The authors show that while there was a notable increase in targets and activities right after the invasion, these activities also ended abruptly (as they began).

Overall, the paper takes an interesting approach to assess how volunteers (or at least self-motivated users) can impact military conflicts in the digital world (i.e., “cyber war”). Our community lacks papers in this direction, and the presented work highlights but also refutes some generally assumed narratives. Thus, the work provided valuable contributions to better understanding the impact of “low-level cybercrime actors” in large-scale conflicts.

The paper is well-written and easy to follow. However, the analysis of the three phenomena (i.e., DDoS, Forums, and defacements) are very similar, and reading the three sections feels somewhat repetitive. It is also worth highlighting that several critical parts of the method are outsourced into the appendix, leading to a state that the paper is not self-contained.

Aside from the fact that most parts of the method are in the appendix, the analysis seems sound and complete. Regarding the selection of the defacement archives, it is not clear how the top five archives were selected. A reference or description of the method and how they were selected would be beneficial.

Generally, most figures and tables are very hard to read in their original size. My main concern with the analysis is that either ANOVA or Kruskal-Willis is used, but it is unclear which test was used in a concrete statistical test (e.g., in Table 2). To this reviewer, it would be more reasonable to just one on tests (Kruskal-Willis) and to report the p-values. That would make the numbers comparable, and it would be clear which test was used.

An exciting addition to the analysis would be to understand if and to what extent new users entered the ecosystem and participated in the attacks and to understand better the target (and potential impact of an attack) of each attack. But given the page limitation, that is probably something that cannot be included.

In conclusion, this reviewer thinks that the paper should be accepted at the conference.

**Response to discussion**
Thank you for the numerous clarifications in the discussion. As they addressed my main concerns (e.g.,data selection processes), I increased my rating of your work.

**Questions:**

Please see the review.

**Ethics Review Description:**

Ethical issues are sufficently adressed.

**Reviewer Confidence:**

4: The reviewer is certain that the evaluation is correct and very familiar with the relevant literature

**Scope:**

4: The work is relevant to the Web and to the track, and is of broad interest to the community

---

### Official Review · Reviewer_PVya · 2023-11-21

**Novelty:** 4
**Technical Quality:** 4

**Review:**

In this paper, the authors have conducted a measurement study on low-level cybercrime actors in the RU conflict. They collected cybercrime related data from multiple different sources including defacement archives and underground forums, and they have analyzed the collected data to understand what sort of cybercrimes happened during RU conflict.

While the paper shows diverse measurement results, it does not deliver new insights or surprising results. Most of analysis results just present quite straightforward or easy to guess findings, which is the most critical problem of this paper.
In the case of the defacement attack result, the authors claim that the distribution of attacks (in terms of the timeline) in UK conflict is statistically different, sudden spikes, from other cases. Indeed, we can easily expect it will happen, and then what are new insights from this measurement? In Figure 1, there are also other sudden spikes, targeting CA or BR, and then what are the differences between those cases and the RU case? The paper simply shows some graphs and narrates their known results, but it does not have any in-depth analysis; why these results are important in understanding the patterns of cybercrimes. Most other results are also quite straightforward (no new insights).

The measurement results on the hacking communities may be interesting, if they have some analysis results on their real contents. For example, the authors may present if there are real discussions on hacking each country. As such, more in-depth analysis is required in most cases.

Overall, the topic selected in the paper is somewhat interesting, but the paper fails in delivering novel ideas or insights.

**Questions:**

- what are the new values in your findings? there are several technical blog aricles or white papers presenting similar issues in public web. what are new things?

- why do we need to care about those low-level cybercrime actors?

**Reviewer Confidence:**

3: The reviewer is confident but not certain that the evaluation is correct

**Scope:**

3: The work is somewhat relevant to the Web and to the track, and is of narrow interest to a sub-community

---

### Official Review · Reviewer_TDSg · 2023-11-22

**Novelty:** 5
**Technical Quality:** 5

**Review:**

The paper examines the engagement of low-level cybercrime actors in the Russia-Ukraine conflict, analyzing a comprehensive dataset consisting of 358k web defacement attacks, 1.7M reflected DDoS attacks, numerous forum posts, and announcements by a volunteer hacking group. The study spans six months, including two months before and four months after the invasion. It concludes that while the conflict initially caught the attention of these actors, resulting in increased online discussions and attacks, their involvement was fleeting and minor compared to more persistent state-sponsored operations​​.

Strengths:
1. The paper leverages an extensive dataset, encompassing various types of cyberattacks and discussions from relevant online forums. This breadth of data provides a robust foundation for the paper's conclusions.
2. The study employs rigorous statistical analysis methods, including One-way ANOVA and Kruskal-Wallis tests, to evaluate the significance of changes in cyberattack patterns across different periods​​.
3. The paper offers valuable insights into the dynamics of low-level cybercrime activities during geopolitical conflicts, highlighting how such actors can be mobilized briefly but are not sustained participants in cyberwarfare​​.
4. Focusing on low-level cybercrime actors in the context of a high-profile geopolitical conflict is a novel approach, filling a gap in cyberwarfare research.

Weaknesses:
1. Narrow Focus on Low-Level Actors: The paper's concentration on low-level cybercrime actors, while novel, might overlook the interconnected nature of cyberwarfare, where low-level activities often intersect with more sophisticated, state-backed operations. This narrow focus could lead to an incomplete understanding of the full cyberwarfare landscape in the Russia-Ukraine conflict.
2. Lack of Comparative Analysis: The study does not compare the role of low-level actors in the Russia-Ukraine conflict with other similar geopolitical situations.
3. Potential Data Incompleteness: While the dataset is extensive, it relies heavily on publicly accessible sources like web defacement archives and forum posts. This reliance might result in a skewed perspective.
4. Lack of Qualitative Insights: The study predominantly uses quantitative data, which limits its ability to delve into the qualitative aspects of cybercrime, such as the motivations and strategies of individual actors. Incorporating qualitative analysis could have enriched the understanding of these actors' roles and impacts.
5. The study covers a period of six months, which, while substantial, might not be long enough to capture the long-term trends and shifts in cybercrime activities related to the conflict.
6. The selection of specific forums and volunteer groups for analysis might introduce bias, as it does not necessarily represent the entire spectrum of low-level cybercrime activities.
7. The use of statistical methods like One-way ANOVA and Kruskal-Wallis tests, though rigorous, might not fully capture the complexity and evolving nature of cyberwarfare activities.
8. Given the dynamic nature of cyberwarfare activities, incorporating time series analysis could provide a more nuanced understanding of the trends and patterns over time.

The paper under review offers valuable insights into the role of low-level cybercrime actors in the Russia-Ukraine conflict, utilizing a substantial dataset of various cyberattacks and online discussions. However, its potential is somewhat limited by the current statistical methodologies employed. A more dynamic approach, incorporating advanced statistical techniques such as time series analysis, network analysis, and predictive modeling, would more accurately reflect the complexities and evolving nature of cyberwarfare. Additionally, integrating qualitative analyses would deepen the understanding of the motivations and strategies of these actors. By expanding its methodological scope, the paper could significantly enhance its contribution to the field, providing a richer, more comprehensive perspective on the intersection of cybercrime and geopolitical conflicts.

**Questions:**

1. Could you provide more details on the selection criteria for the data sources used, particularly the web defacement archives and online forums? How did you ensure that these sources were representative of the broader spectrum of cybercrime activities during the conflict?
2. What was the rationale behind choosing One-way ANOVA and Kruskal-Wallis tests for your analysis? Did you consider the use of time series analysis or other statistical methods that account for temporal dependencies and dynamic changes in cyberwarfare activities?
3. Given the reliance on publicly accessible data, how did you address the potential bias towards more visible or reported cyber activities? Were there any measures taken to account for or mitigate this bias in your analysis?
4. The study covers a six-month period. How do you believe the trends and patterns observed in this period would evolve over a longer timeframe? Would a longer analysis period potentially alter your conclusions?
5. Could you elaborate on the implications of your findings for international cybersecurity policies and strategies? How might your insights inform the development of more effective cyber defense mechanisms?

**Reviewer Confidence:**

4: The reviewer is certain that the evaluation is correct and very familiar with the relevant literature

**Scope:**

4: The work is relevant to the Web and to the track, and is of broad interest to the community

---

### Decision · Program_Chairs · 2024-01-22

**Decision:**

Accept (Oral)

**Comment:**

**Meta Review:**

 **Pros:**

 1. **Comprehensive Data Analysis:** Reviewers appreciate the paper's in-depth analysis of low-level cybercrime actors' engagement in the Russia-Ukraine conflict using a comprehensive dataset. The inclusion of various cyberattack types and discussions from forums adds richness to the study.

 2. **Statistical Rigor:** The use of statistical methods, such as One-way ANOVA and Kruskal-Wallis tests, is seen as a strength, providing a quantitative basis for evaluating the significance of changes in cyberattack patterns across different periods.

 3. **Novel Approach:** The paper is commended for its novel approach in focusing on low-level cybercrime actors during a high-profile geopolitical conflict. This approach fills a gap in cyberwarfare research and contributes to a better understanding of the dynamics of such actors in the context of a major international conflict.

 4. **Clear Presentation:** The paper is well-written and easy to follow, enhancing its accessibility and comprehension.

 **Cons:**

 1. **Narrow Focus on Low-Level Actors:** Some reviewers express concern about the paper's narrow focus on low-level cybercrime actors, suggesting that it might overlook the broader context where low-level activities intersect with more sophisticated, state-backed operations.

 2. **Lack of Comparative Analysis:** The absence of a comparative analysis with other geopolitical situations is noted as a weakness. Comparing the role of low-level actors in the Russia-Ukraine conflict with similar conflicts could provide additional insights.

 3. **Potential Data Incompleteness:** There are concerns about the potential bias introduced by relying on publicly accessible sources, which might result in a skewed perspective. Addressing this bias and ensuring representativity of the dataset is considered important.

 4. **Lack of Qualitative Insights:** The predominance of quantitative data limits the study's ability to delve into qualitative aspects, such as motivations and strategies of individual actors. Incorporating qualitative analysis is suggested to enrich the understanding of these actors' roles.

 5. **Limited Analysis Period:** The six-month analysis period is noted as a potential limitation. Reviewers suggest considering how trends and patterns might evolve over a longer timeframe for a more comprehensive understanding.

 6. **Selection Bias in Forums and Groups:** There's a concern about potential bias introduced by selecting specific forums and volunteer groups for analysis, as they may not represent the entire spectrum of low-level cybercrime activities.

 **Suggestions and Questions:**

 1. **Expand Comparative Analysis:** Reviewers suggest expanding the analysis by comparing the role of low-level actors in the Russia-Ukraine conflict with other geopolitical situations, providing a broader context.

 2. **Qualitative Analysis:** Incorporating qualitative analysis alongside quantitative data could provide a more comprehensive understanding of the motivations and strategies of low-level cybercrime actors.

 3. **Addressing Data Bias:** Clarifying measures taken to address potential bias introduced by publicly accessible data and ensuring the representativity of the dataset is crucial.

 4. **Extended Analysis Period:** Considering how trends and patterns might evolve over a longer timeframe could strengthen the study's insights into the dynamics of low-level cybercrime during geopolitical conflicts.

 5. **International Cybersecurity Implications:** Exploring the implications of findings for international cybersecurity policies and strategies would enhance the paper's relevance and contribution to the field.

 **Conclusion:**

 The paper is acknowledged for its comprehensive data analysis and statistical rigor in exploring the role of low-level cybercrime actors in the Russia-Ukraine conflict. However, addressing potential biases, incorporating qualitative analysis, expanding the comparative context, and considering advanced statistical techniques are crucial for enhancing the paper's contribution and relevance. Balancing the focus on low-level actors with an awareness of the broader cyberwarfare landscape is essential for a more complete understanding of the subject.

 ---